# Identification of two novel adenoviruses in smooth-billed ani and tropical screech owl

**Ana Paula Jejesky de Oliveira**[1]*, **Maria Cristina Valdetaro Rangel**[1], **Márton Z. Vidovszky**[2], **João Luiz Rossi, Jr**[1], **Fernando Vicentini**[3], **Balázs Harrach**[2], **Győző L. Kaján**[2]

**1** Laboratory of Wildlife Health, Department of Ecosystem Ecology, University of Vila Velha, Vila Velha, ES, Brazil, **2** Institute for Veterinary Medical Research, Centre for Agricultural Research, Budapest, Hungary, **3** Health Sciences Center, Federal University of Recôncavo da Bahia, Santo Antônio de Jesus, BA, Brazil

* anapaulaoliveira799@yahoo.com.br

## Abstract

Avian adenoviruses (AdVs) are a very diverse group of pathogens causing diseases in poultry and wild birds. Wild birds, endangered by habitat loss and habitat fragmentation in the tropical forests, are recognised to play a role in the transmission of various AdVs. In this study, two novel, hitherto unknown AdVs were described from faecal samples of smooth-billed ani and tropical screech owl. The former was classified into genus *Aviadenovirus*, the latter into genus *Atadenovirus*, and both viruses most probably represent new AdV species as well. These results show that there is very limited information about the biodiversity of AdVs in tropical wild birds, though viruses might have a major effect on the population of their hosts or endanger even domesticated animals. Surveys like this provide new insights into the diversity, evolution, host variety, and distribution of avian AdVs.

**Data Availability Statement:** Sequences were submitted to the NCBI GenBank under accession numbers MN540447 and MN540448.

## Introduction

Urbanisation, land conversion and road-building pose a risk to wildlife by habitat fragmentation and also by exposing the animals to the risk of roadkill [1]. Habitat loss and fragmentation modify the landscape and have consequences for biodiversity conservation [2,3]. The highway ES-060 in Brazil crosses three important environmental reserves: the Jacarenema Ecological Reserve (307 ha), the Setiba Environmental Protection Area (12,960 ha) and the Paulo César Vinha State Park (1,500 ha). The vegetation of these reserves is mainly composed of restinga, a distinct type of tropical moist broadleaf forest, regarded as a key area for biodiversity [4]. The rural expansion is intensive adjacent to the highway.

Among the negative anthropic actions, the establishment of new roads is among the most impacting changes of environment [5]. Beyond the risk of roadkill, roads inhibit the movement of many species and act as barriers (total or partial), isolating populations [2,3], with few species immune to this threat [6]. The likelihood of roadkill depends on the animal movement patterns and the landscape characteristics [7,8]. Animal movement patterns are the result of behavioural trade-offs, and are influenced by the individual's internal state and also by the environment [9]. When roads or railroads cross movement routes, this creates areas of greater risk for the wildlife, for human safety and also for domestic animals [7].

**Funding:** The research of APJO is supported by the Espírito Santo Research and Innovation Support Foundation (438/2016, fapes.es.gov.br) and University of Vila Velha (201672442, www.uvv.br). GLK is supported by the OMA Foundation (101öu6, omaa.hu), and he is also the recipient of the János Bolyai Research Scholarship of the Hungarian Academy of Sciences (https://mta.hu/english). The research of MZV and BH is supported by the National Research, Development and Innovation Office (NN128309, nkfih.gov.hu/english-nkfih).

**Competing interests:** The authors have declared that no competing interests exist.

Adenoviruses (AdVs) are DNA viruses with an icosahedral capsid and a double-stranded, linear genome. The presence of AdVs is described in many species of vertebrate animals, including mammals, birds, reptiles, amphibians, and fish [10–14]. It is known that birds are common hosts for AdVs, a fact that has been mirrored by the large number of avian AdVs [15–25]. The host bird species often live in crowded flocks, and migrate large distances by flying, and this predisposes to a rapid viral dissemination in the environment.

The International Committee on Taxonomy of Viruses recognises five genera belonging to the family *Adenoviridae*. Birds can be infected by highly divergent AdVs classified into three different genera: genus *Aviadenovirus*, *Siadenovirus* and *Atadenovirus* [11]. The pathogenicity of these viruses is not always clear, they can cause latent infections but diseases as well, depending on the virulence of the strain and also on cospeciation time [26]. The deeper evolutionary history of the family *Adenoviridae* still needs to be resolved, and the discovery of new AdVs in new hosts provides more accurate phylogenetic trees and better understanding of the co-evolution and host switches of these viruses.

Wild birds play an important role in the transmission and as reservoirs of various AdVs [27–35]. Habitat loss and fragmentation cause novel interactions between pathogens, hosts and the environment. This creates new routes for disease transmission, which results in the possible dispersion and adaptation of pathogens to new hosts. To address this risk, we investigated the occurrence of AdVs in tropical screech owls (*Megascops choliba*), guira cuckoos (*Guira guira*) and smooth-billed anis (*Crotophaga ani*) found dead along the highway ES-060.

## Materials and methods

### 1. Study area

The ES-060 highway crosses the municipalities of Vitória, Vila Velha and Guarapari. The northern end of the highway is located at the coordinates 20˚18'48.50" S– 40˚17'32.48" W, and the southern at the coordinates 21˚18'5.44" S– 41˚0'7.77" W.

### 2. Origin of samples

The highway is monitored every 90 minutes for accidents or roadkill occurrence. The animals found alive or dead are collected, recorded and transferred for veterinary treatment or disposal. Faecal samples were collected from 19 dead birds in 2017: five tropical screech owls, four guira cuckoos and ten smooth-billed anis. Samples were collected during necropsies directly from the rectal ampulla of the birds.

For faecal suspension, approximately 200 mg of faecal sample was diluted to a concentration of 20% in Tris-calcium buffer (Tris 0.01 M, $CaCl_2$ 1.5 mM, pH 7.2), homogenised and centrifuged at 2000 *g* for 10 min. Nucleic acid was extracted by the method of Boom et al. [36], using guanidine thiocyanate and silica particles (product numbers: 50983 and 107536, respectively, both from Merck, Darmstadt, Germany).

### 3. PCR and sequencing

Extracted DNA samples were screened for the presence of AdVs using a pan-adenovirus PCR, targeting the gene of the viral DNA polymerase and detecting all known AdVs [10,37]. The 321-bp-long PCR products were purified using the NucleoSpin Gel and PCR Clean-up Kit (Macherey-Nagel; Düren, Germany), and sequenced using the BigDye Terminator v3.1 Cycle Sequencing Kit (Thermo Fisher Scientific; Waltham, Massachusetts, United States of America) on both strands according to the manufacturers' protocol.

## 4. Phylogenetic analysis

The acquired sequences were assembled and translated to amino acid sequences using Geneious 9.1.8. For phylogenetic tree inference, the multiple alignment was conducted using MAFFT [38], the length of the multiple alignment was 90 amino acids. The evolutionary model selection and the phylogenetic calculation were performed using RAxML 8.2.10 [39], the best-scoring model was LG with empirical base frequencies. The robustness of the tree was determined with a non-parametric bootstrap calculation using 1,000 repeats. The phylogenetic tree was visualised using MEGA 7 [40], the tree was rooted on the midpoint, and bootstrap values were given as percentages if they reached 75%. The obtained sequences were compared to entries of the NCBI Protein database using BlastX [41] on 20/09/2019.

## Results and discussion

PCR products were gained from three smooth-billed ani samples and one tropical screech owl sample. The GenBank accession numbers for the sequences are MN540447 and MN540448.

The result of the phylogenetic analysis is displayed in Fig 1. The three smooth-billed ani AdV strains shared identical nucleic acid sequences on the conserved stretch of the DNA polymerase, the virus strain clustered into genus *Aviadenovirus*. The BlastX hit with the highest pairwise amino acid sequence identity was pigeon AdV-2 with 75.6% (E value: $4.22 \times 10^{-43}$, query coverage: 100%). The screech owl AdV clustered into genus *Atadenovirus*, and shared 80.0% sequence identity with the helodermatid AdV-1 (syn. lizard AdV-1 or gila monster AdV; E value: $2.31 \times 10^{-47}$, query coverage: 100%). This level of sequence divergence suggests that these virus strains represent novel AdV species, as the primary species demarcation criterion for AdVs is 10–15% sequence divergence on the DNA-dependent DNA polymerase amino acid sequence [11]. However, further description of the virus is required to support this hypothesis, primarily, more genomic information—including the complete sequence of the DNA polymerase—would be essential.

As identical virus sequences were detected in the three smooth-billed anis, the identified AdV is most probably infectious for this bird species, and it is not a foodborne contaminant. A long-term co-evolution is hypothesised between this aviadenovirus and its host as aviadenoviruses infect a wide range of bird species [12,27,31–34,42]. Though infectious, it may not be evidently pathogenic, as co-evolving viruses are often non-pathogenic or facultatively pathogenic to their host species [26,43–46]. No pathological findings were observed apart from trauma due to vehicular damage neither in these anis nor in the screech owl.

The tropical screech owl AdV was detected in one bird only, and clustered into genus *Atadenovirus*. Atadenoviruses are thought to have coevolved with squamate reptiles [47,48], but several bird species are also infected by atadenoviruses [17,23,49]. As this owl species is a bird of prey, it is equally possible that this virus had already adapted to this host and replicated in this bird, or that this virus is a foodborne contaminant. Tropical screech owls are omnivorous and known to prey on small reptiles, rodents, and amphibians [50], so the detected atadenovirus might originate from a prey reptile. If the latter will be supported by future results, the designation tropical screech owl associated AdV is recommended for the strain.

The tropical screech owl (family Strigidae) and the smooth-billed ani (family Cuculidae) are species widely distributed in Brazil but also endemic from Costa Rica to Paraguay and northern Argentina [51,52]. The two species of birds act directly on the population dynamics of the prey populations and are able to contribute to maintaining the diversity of these communities [53,54] and to produce secondary effects in plant communities [55].

These samples show us that there is very limited information about the biodiversity of AdVs in tropical wild birds, though viruses might have a major effect on the population of

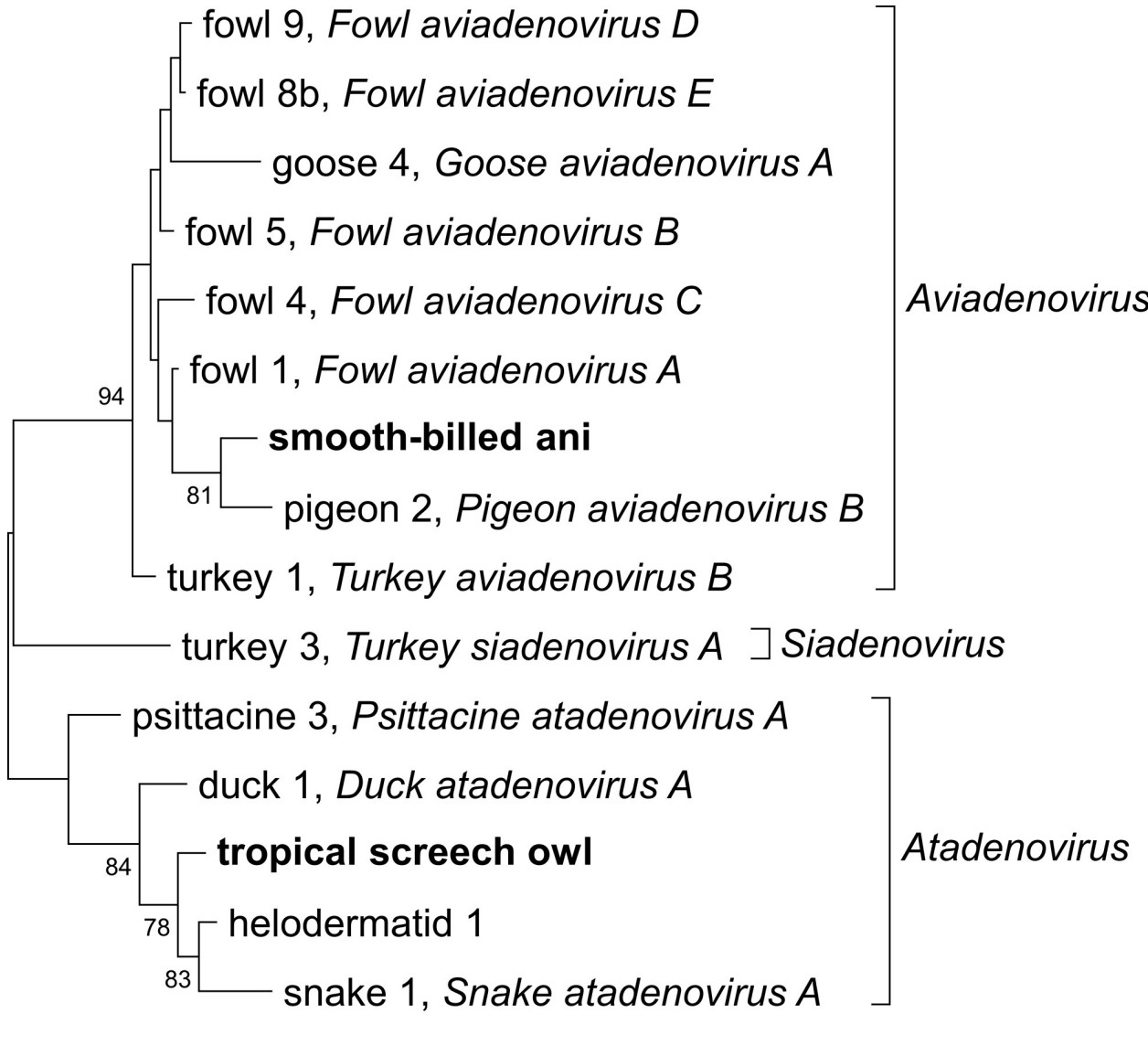

**Fig 1. Phylogenetic analysis of the smooth-billed ani and the tropical screech owl adenoviruses.** The tree is based on derived amino acid sequence of partial DNA polymerase gene sequences. Adenovirus strains are represented using the host name and the serotype ordinal number, and viral species names are also applied if available. The tree was rooted on the midpoint. Accession numbers: duck 1: NP_044702, fowl 1: AAC54904, fowl 4: AEK64762, fowl 5: YP_007985646, fowl 8b: ANJ02558, fowl 9: derived from AC_000013, goose 4: YP_006383556, helodermatid 1: AAS89696, pigeon 2: APO40944, psittacine 3: YP_009112716, smooth-billed ani: QHA24662, snake 1: YP_001552247, tropical screech owl: QHA24661, turkey 1: YP_003933581, turkey 3: NP_047384.

their hosts or endanger even domesticated animals. Several studies have already been published about poultry AdVs [56,57], but further investigations are required to shed light on the diversity and pathogenicity of AdVs in tropical host animals.

Rural expansion causes irreversible damages to wildlife, all 19 birds investigated were found dead because of road traffic accidents. As human activity intensifies, an increasing number of pet animals and livestock are raised in the close proximity of previously unharmed natural habitats. This enhances the risk of viral spread and host changes, but this phenomena was not

observed in this investigation: the detected virus strains are not of domestic animal origin, nor have been found in domesticated or pet animals yet.

Due to the ability of avian AdVs—mainly aviadenoviruses—to cause subclinical infections, the diversity of AdVs in birds is probably much more extensive than thought before [17]. The identification of two novel AdVs in specimens of two Brazilian tropical bird species suggests that numerous further unknown AdVs are circulating in other tropical bird species. Surveys like this provide new insights into the pathogenicity, diversity, evolution, host variety, and distribution of avian AdVs; thus, similar studies should be conducted in different geographical locations too.

## Acknowledgments

The authors would like to thank the concessionaire of the Rodovia do Sol System and the Sinhá Laurinha Society.

## Author Contributions

**Conceptualization:** Ana Paula Jejesky de Oliveira, Fernando Vicentini.

**Formal analysis:** Márton Z. Vidovszky, Balázs Harrach, Győző L. Kaján.

**Funding acquisition:** João Luiz Rossi, Jr.

**Methodology:** Ana Paula Jejesky de Oliveira, Maria Cristina Valdetaro Rangel, Fernando Vicentini.

**Project administration:** Ana Paula Jejesky de Oliveira.

**Supervision:** João Luiz Rossi, Jr, Fernando Vicentini.

**Writing – original draft:** Ana Paula Jejesky de Oliveira, Márton Z. Vidovszky, Fernando Vicentini, Balázs Harrach, Győző L. Kaján.

**Writing – review & editing:** Ana Paula Jejesky de Oliveira, Márton Z. Vidovszky, Fernando Vicentini, Balázs Harrach, Győző L. Kaján.

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
