## [Decision Letter · Decision Letter 0]

18 Dec 2019

PONE-D-19-31175

Identification of two novel adenoviruses in smooth-billed ani and tropical screech owl

PLOS ONE

Dear mrs JEJESKY DE OLIVEIRA,

Thank you for submitting your manuscript to PLOS ONE. After careful consideration, we feel that it has merit but does not fully meet PLOS ONE’s publication criteria as it currently stands. Therefore, we invite you to submit a revised version of the manuscript that addresses the points raised during the review process.

Please address all reviewer concerns as described below prior to submission. Also, provide full sequences and a web link to the Genebank sequences.

We would appreciate receiving your revised manuscript by Feb 01 2020 11:59PM. To enhance the reproducibility of your results, we recommend that if applicable you deposit your laboratory protocols in protocols.io, where a protocol can be assigned its own identifier (DOI) such that it can be cited independently in the future. For instructions see: http://journals.plos.org/plosone/s/submission-guidelines#loc-laboratory-protocols

We look forward to receiving your revised manuscript.

Kind regards,

Negin P. Martin, Ph.D.

Academic Editor

PLOS ONE

Journal Requirements:

2. We note that you are reporting an analysis of a microarray, next-generation sequencing, or deep sequencing data set. PLOS requires that authors comply with field-specific standards for preparation, recording, and deposition of data in repositories appropriate to their field. Please upload these data to a stable, public repository (such as ArrayExpress, Gene Expression Omnibus (GEO), DNA Data Bank of Japan (DDBJ), NCBI GenBank, NCBI Sequence Read Archive, or EMBL Nucleotide Sequence Database (ENA)). In your revised cover letter, please provide the relevant accession numbers that may be used to access these data. For a full list of recommended repositories, see http://journals.plos.org/plosone/s/data-availability#loc-omics or http://journals.plos.org/plosone/s/data-availability#loc-sequencing.

Reviewers' comments:

Reviewer's Responses to Questions

**Comments to the Author**

1. Is the manuscript technically sound, and do the data support the conclusions?

Reviewer #1: Partly

Reviewer #2: Yes

Reviewer #3: Partly

Reviewer #4: Yes

2. Has the statistical analysis been performed appropriately and rigorously? 

Reviewer #1: N/A

Reviewer #2: N/A

Reviewer #3: N/A

Reviewer #4: N/A

3. Have the authors made all data underlying the findings in their manuscript fully available?

Reviewer #1: Yes

Reviewer #2: Yes

Reviewer #3: No

Reviewer #4: Yes

4. Is the manuscript presented in an intelligible fashion and written in standard English?

Reviewer #1: Yes

Reviewer #2: Yes

Reviewer #3: No

Reviewer #4: Yes

5. Review Comments to the Author

Reviewer #1: The MS titled “Identification of two novel adenoviruses in smooth-billed ani and tropical screech owl” describes the amplification of a DNA polymerase domain from fecal samples from 19 raod kill animals. The amplicon sequences (translates to ~90 aa sequence) is used to determine that there are two new species of adenoviruses associated with these birds. I’ll be honest that I find it strange that a 90aa sequence can be used on this day and age for typing viruses to a species level.

I have highlighted some comments that I think are important to address

Line 30-32: this is a rather odd introduction to a manuscript.

Line 86-90: provide details of the size of the amplicon as the reader is aware that the MS is bases on a small region of the DNA polymerase. Also, if you undertook direct sequencing of PCR products, I am assuming you should have had slightly noisy chromatograms due to polymorphisms.

Line 94-93: RAxML does not do model selection. It is a maximum likelihood inference software. You would have had to use ProtTest to determine the best model. Please provide the best fit model.

Line 104: Two accession #s but you have more than two positives?

Line 107-110: This BLASTx data. You need to provide coverage and e-value. The coverage may be 10% with 75.6% identity.

Reviewer #2: Avian adenoviruses (AdVs) are a very diverse group of pathogens causing the diseases in poultry and wild birds. In this study, two novel AdVs were described from faecal samples of smooth-billed and tropical screech owl. District type of tropical broadleaf forest regarded area for biodiversity. The former was classified into genus Aviadenovirus, the latter into genus Atadenovirus, and both viruses most probably represent new AdV species. These results show that there is very limited information about the biodiversity of AdVs in tropical wild birds, and the diversity of these viruses is probably much more extensive than thought before. This viruses might have a major effect on the population of their hosts or endanger even domesticated animals. Surveys like this provide new insights into the diversity, evolution, host variety, and distribution of AdVs.

As authors pointed out the surveys like this provide new insights into the pathogenicity, diversity, evolution, host variety, and distribution of avian AdVs.

The manuscript is important from epidemiological point of view and I am for the publication of this manuscript in Plos One, however, the proofreading is needed.

Reviewer #3: The manuscript having title "Identification of two novel adenoviruses in smooth-billed ani and tropical screech owl" needs to be improved as it requires major revisions. Moreover, the manuscript does not describe significant novel findings. So, i think it is not suitable for the journal like "PLOS ONE".

English language should be improved and it should be revised by a native English speaker.

Title of the manuscript is not appropriate, it may be changed to "Molecular characterization of adenoviruses associated with smooth-billed ani and tropical screech owl".

Reviewer #4: The manuscript describes the detection and identification of avian adenoviruses in wild birds native in the tropical rain forests of Brazil. The manuscript is well written and provides new insights into the diversity of AAdVs.

However, the following comments should be addressed:

@abstract:

L27: avian AdVs

@ introduction:

The issue of urbanisation, land conversion, roads etc. having significant effect on wildlife populations and even the distribution of pathogens, are mentioned extensively in the introduction (L36-47). However, the discussion in this regard is lacking and in parts contradictory to the introduction/hypothesis (new hosts 'L66' - long term co-evolution of virus and host 'L123-127').

In L64-66 authors describe screech owls and ani as new hosts. To my knowledge these birds were investigated for the first time for the prevalence of AdVs - how do authors define 'new' hosts?

@ Material and methods:

L75-80 - origin of samples - what was the timeframe in which samples were procured?

L81-90 - please complete information in regards to manufacturers (e.g. silica particles, Clean-up kit..)

@Figure 1:

information on gene sequences used should include accession numbers (and possibly species or country of origin) in order to facilitate comparative analyses.

@Discussion:

L123 - Please add information on nutritional habits of ani (granivor, carnivor??) if you mention possibility of foodborne contaminants.

L126-127 - Was necropsy (mentioned in L79-80) indicative for any pathological changes apart from trauma due to vehicular damage?

@References:

With respect to the length of the manuscript 57 references seem excessive? Are all references necessary?

6. PLOS authors have the option to publish the peer review history of their article (what does this mean?). If published, this will include your full peer review and any attached files.

Reviewer #1: No

Reviewer #2: No

Reviewer #3: Yes: Dr. M. Salahuddin Shah

Reviewer #4: No

---

## [Author Response · Author response to Decision Letter 0]

27 Jan 2020

Negin P. Martin, Ph.D.

Academic Editor

PLOS ONE

January 27, 2020

Subject: rebuttal letter

Dear Dr. Martin,

The authors are grateful for the reviewers for their thorough work. The manuscript was heavily edited to meet the recommendations of the reviewers. Please find our answers below. We hope you can find these and the revised version acceptable. 

Sincerely,

Ana Paula Jejesky de Oliveira

Corresponding author

Journal Requirements:

The MS was edited to meet the style requirements.

2. We note that you are reporting an analysis of a microarray, next-generation sequencing, or deep sequencing data set. PLOS requires that authors comply with field-specific standards for preparation, recording, and deposition of data in repositories appropriate to their field. Please upload these data to a stable, public repository (such as ArrayExpress, Gene Expression Omnibus (GEO), DNA Data Bank of Japan (DDBJ), NCBI GenBank, NCBI Sequence Read Archive, or EMBL Nucleotide Sequence Database (ENA)). In your revised cover letter, please provide the relevant accession numbers that may be used to access these data. For a full list of recommended repositories, see http://journals.plos.org/plosone/s/data-availability#loc-omics or http://journals.plos.org/plosone/s/data-availability#loc-sequencing.

Sequences were submitted to the NCBI GenBank under accession numbers MN540447 and MN540448. The sequences are released now.

 

Reviewers' comments:

Reviewer's Responses to Questions

Comments to the Author

1. Is the manuscript technically sound, and do the data support the conclusions?

Reviewer #1: Partly

Reviewer #2: Yes

Reviewer #3: Partly

Reviewer #4: Yes

The MS was edited heavily to meet the requirements of the Reviewers.

2. Has the statistical analysis been performed appropriately and rigorously?

Reviewer #1: N/A

Reviewer #2: N/A

Reviewer #3: N/A

Reviewer #4: N/A

3. Have the authors made all data underlying the findings in their manuscript fully available?

Reviewer #1: Yes

Reviewer #2: Yes

Reviewer #3: No

Reviewer #4: Yes

Sequences were submitted to the NCBI GenBank under accession numbers MN540447 and MN540448. The sequences are released now.

4. Is the manuscript presented in an intelligible fashion and written in standard English?

Reviewer #1: Yes

Reviewer #2: Yes

Reviewer #3: No

Reviewer #4: Yes

The MS was edited heavily to enhance its readability.

5. Review Comments to the Author

Reviewer #1: 

The MS titled “Identification of two novel adenoviruses in smooth-billed ani and tropical screech owl” describes the amplification of a DNA polymerase domain from fecal samples from 19 raod kill animals. The amplicon sequences (translates to ~90 aa sequence) is used to determine that there are two new species of adenoviruses associated with these birds. I’ll be honest that I find it strange that a 90aa sequence can be used on this day and age for typing viruses to a species level.

The reviewer is absolutely right, that species designation would be more accurate based on complete genome data. Furthermore, official proposal of a new adenovirus species by the International Committee on Taxonomy of Viruses would only be possible based on complete genome analysis. 

However, the viruses were detected from fecal samples. Without virus isolation, complete genome sequencing can be challenging from fecal samples. And the isolation would require most possibly primary cells from tropical screech owl and smooth-billed ani embryos. This technique is often used for fowl (chicken) adenoviruses: these viruses are isolated regularly on primary cells originating from killed chicken embryos [1]. Unfortunately, embrionated eggs of these wild birds are not available, but even if they were, their use would be ethically disputable.

Thus, we conducted merely a preliminary prediction about the species of the strains based on their high divergence from other adenoviruses. We have formulated our hypothesis accordingly at l. 117–123. But it must be stressed that according to the International Committee on Taxonomy of Viruses, species designation of adenoviruses depends primarily on the DNA polymerase-based phylogenetic distance [2]. So even with a complete genome sequence in hand, one would use primarily the DNA-polymerase sequence to determine the species of the strain.

I have highlighted some comments that I think are important to address

Line 30-32: this is a rather odd introduction to a manuscript.

The Reviewer is right; the MS was updated at this location (l. 32–37)

Line 86-90: provide details of the size of the amplicon as the reader is aware that the MS is bases on a small region of the DNA polymerase.

The Reviewer is right, amplicon size was missing from the MS, but it was added now (l. 91).

Also, if you undertook direct sequencing of PCR products, I am assuming you should have had slightly noisy chromatograms due to polymorphisms.

The Reviewer is right, animals are often infected by several adenovirus strains causing a sequence polymorphism. In such cases, the molecular cloning of the PCR product is conducted [3]. However, this was not observed in this study, all sequence reads had nice quality, the mean base call confidence values in the four assemblies (originating from three anis and one owl) ranged 45.3–46.4, and the three different smooth-billed ani adenovirus strains shared an identical nucleic acid sequence (l. 111–113).

Line 94-93: RAxML does not do model selection. It is a maximum likelihood inference software. You would have had to use ProtTest to determine the best model. Please provide the best fit model.

The Reviewer is right, RAxML 8 is a maximum likelihood inference software, but it is also capable of determining the protein substitution model using the -m PROTGAMMAAUTO option (manual p. 31). The best-scoring model was LG with empirical base frequencies in our case. This information was included into the MS now (l. 101).

Line 104: Two accession #s but you have more than two positives?

As stated in the MS (l. 111–113), "the three smooth-billed ani AdV strains shared identical nucleic acid sequences". Thus one sequence was deposited for the smooth-billed ani AdV, and one for the tropical screech owl AdV.

Line 107-110: This BLASTx data. You need to provide coverage and e-value. The coverage may be 10% with 75.6% identity.

The query coverages were 100% for both BlastX searches, the E values were 4.22 x 10-43 and 2.31 x 10-47 for the smooth-billed ani and screech owl AdV, respectively. This information was included into the MS now (l. 114-117).

Reviewer #2: 

Avian adenoviruses (AdVs) are a very diverse group of pathogens causing the diseases in poultry and wild birds. In this study, two novel AdVs were described from faecal samples of smooth-billed and tropical screech owl. District type of tropical broadleaf forest regarded area for biodiversity. The former was classified into genus Aviadenovirus, the latter into genus Atadenovirus, and both viruses most probably represent new AdV species. These results show that there is very limited information about the biodiversity of AdVs in tropical wild birds, and the diversity of these viruses is probably much more extensive than thought before. This viruses might have a major effect on the population of their hosts or endanger even domesticated animals. Surveys like this provide new insights into the diversity, evolution, host variety, and distribution of AdVs.

As authors pointed out the surveys like this provide new insights into the pathogenicity, diversity, evolution, host variety, and distribution of avian AdVs.

The manuscript is important from epidemiological point of view and I am for the publication of this manuscript in Plos One, however, the proofreading is needed.

The authors are grateful for the time and effort of the Reviewer. The MS was edited to meet the requirements of the Reviewers and the journal.

Reviewer #3: 

The manuscript having title "Identification of two novel adenoviruses in smooth-billed ani and tropical screech owl" needs to be improved as it requires major revisions. Moreover, the manuscript does not describe significant novel findings. So, i think it is not suitable for the journal like "PLOS ONE".

The MS was edited to enhance its readability and meet the requirements of the Reviewers and the journal. The first criterion for publication in PLOS One is that "the study presents the results of original research". This and all other criteria are met by this MS.

English language should be improved and it should be revised by a native English speaker.

Title of the manuscript is not appropriate, it may be changed to "Molecular characterization of adenoviruses associated with smooth-billed ani and tropical screech owl".

For a molecular characterization the complete genome sequence of the virus strains would be essential. Unfortunately, this is not available to the authors.

Reviewer #4: 

The manuscript describes the detection and identification of avian adenoviruses in wild birds native in the tropical rain forests of Brazil. The manuscript is well written and provides new insights into the diversity of AAdVs.

However, the following comments should be addressed:

@abstract:

L27: avian AdVs

The Reviewer is right, the MS was edited accordingly.

@ introduction:

The issue of urbanisation, land conversion, roads etc. having significant effect on wildlife populations and even the distribution of pathogens, are mentioned extensively in the introduction (L36-47). However, the discussion in this regard is lacking and in parts contradictory to the introduction/hypothesis (new hosts 'L66' - long term co-evolution of virus and host 'L123-127').

The Reviewer is right, the MS was complemented at l. 163–168.

In L64-66 authors describe screech owls and ani as new hosts. To my knowledge these birds were investigated for the first time for the prevalence of AdVs - how do authors define 'new' hosts?

The reviewer is right, the MS was easily misinterpretable at this location.

The word "novel" or "new" has been used in three different contexts in the MS.

1. At l. 64–66 (now l. 66–68) the authors tried to express the possibility of host switches due to environmental changes. Such changes might provide new hosts for the virus strains.

2. Adenoviruses were detected in the host species tropical screech owl and smooth-billed ani for the first time.

3. These two viral strains might represent two novel, hitherto undescribed AdV species as well, not accepted by the International Committee on Taxonomy of Viruses yet.

To enhance readability and clarify the MS, it was edited at l. 22, 25, 68 and 166–168.

@ Material and methods:

L75-80 - origin of samples - what was the timeframe in which samples were procured?

Samples have been collected for a year during 2017. This information was included into the MS now (l. 80).

L81-90 - please complete information in regards to manufacturers (e.g. silica particles, Clean-up kit..)

The Reviewer is right, information about the manufacturers were complemented at l. 86–94.

@Figure 1:

information on gene sequences used should include accession numbers (and possibly species or country of origin) in order to facilitate comparative analyses.

The Reviewer is right, accession numbers are included into the figure legend (l. 129–134) as well as viral species names into Figure 1.

@Discussion:

L123 - Please add information on nutritional habits of ani (granivor, carnivor??) if you mention possibility of foodborne contaminants.

The Reviewer is right, this information is included into the MS now (l. 149–152).

L126-127 - Was necropsy (mentioned in L79-80) indicative for any pathological changes apart from trauma due to vehicular damage?

No other pathological findings were observed, and this information is included into the MS now (l. 141–143).

@References:

With respect to the length of the manuscript 57 references seem excessive? Are all references necessary?

There are no limitations by PLOS One for the number of references, and articles are not printed, so no environmental questions should be considered. Thus, the authors decided to give credit where credit is due, and also provide appropriate references to mentioned issues and applied techniques. Especially in the case of wild bird adenoviruses, these references serve as a collection of recommended literature as well.

References

1. Kaján GL, Affranio I, Tóthné Bistyák A, Kecskeméti S, Benkő M. An emerging new fowl adenovirus genotype. Heliyon. 2019;5: e01732. doi:10.1016/j.heliyon.2019.e01732

2. Harrach B, Benkő M, Both GW, Brown M, Davison AJ, Echavarría M, et al. Family Adenoviridae. In: King AMQ, Lefkowitz E, Adams MJ, Carstens EB, editors. Virus Taxonomy: IXth Report of the International Committee on Taxonomy of Viruses. San Diego: Elsevier; 2011. pp. 125–141. 

3. Kaján GL, Kecskeméti S, Harrach B, Benkő M. Molecular typing of fowl adenoviruses, isolated in Hungary recently, reveals high diversity. Vet Microbiol. 2013;167: 357–363. doi:10.1016/j.vetmic.2013.09.025

---

## [Editor Report · Decision Letter 1]

6 Feb 2020

Identification of two novel adenoviruses in smooth-billed ani and tropical screech owl

PONE-D-19-31175R1

Dear Dr. JEJESKY DE OLIVEIRA,

We are pleased to inform you that your manuscript has been judged scientifically suitable for publication and will be formally accepted for publication once it complies with all outstanding technical requirements.

With kind regards,

Negin P. Martin, Ph.D.

Academic Editor

PLOS ONE
---

## [Editor Report · Acceptance letter]

11 Feb 2020

PONE-D-19-31175R1 

Identification of two novel adenoviruses in smooth-billed ani and tropical screech owl 

Dear Dr. Jejesky de Oliveira:

I am pleased to inform you that your manuscript has been deemed suitable for publication in PLOS ONE. Congratulations! Your manuscript is now with our production department. 

With kind regards,

on behalf of

Dr. Negin P. Martin 

Academic Editor

PLOS ONE